# A Graph-Based Hybrid Reconfiguration Deformation Planning for Modular Robots

**DOI:** 10.3390/s23187892

**Published:** 2023-09-14

**Authors:** Ruopeng Wei, Yubin Liu, Huijuan Dong, Yanhe Zhu, Jie Zhao

**Affiliations:** State Key Laboratory of Robotics and Systems, Harbin Institute of Technology, Harbin 150001, China; weiruopeng_hit@163.com (R.W.);

**Keywords:** modular robotics, self-reconfiguration, mobile robotics, reconfiguration deformation, swarm robotics, path planning for multiple mobile robots

## Abstract

The self-reconfigurable modular robotic system is a class of robots that can alter its configuration by rearranging the connectivity of their component modular units. The reconfiguration deformation planning problem is to find a sequence of reconfiguration actions to transform one reconfiguration into another. In this paper, a hybrid reconfiguration deformation planning algorithm for modular robots is presented to enable reconfiguration between initial and goal configurations. A hybrid algorithm is developed to decompose the configuration into subconfigurations with maximum commonality and implement distributed dynamic mapping of free vertices. The module mapping relationship between the initial and target configurations is then utilized to generate reconfiguration actions. Simulation and experiment results verify the effectiveness of the proposed algorithm.

## 1. Introduction

Self-reconfigurable modular robots represent a novel class of robots that can autonomously change their own shape by rearranging the connectivity of their component parts [1]. This ability to dynamically reconfigure opens up new possibilities for adaptability and versatility in robot systems [2]. A fundamental problem to achieve the ability is the reconfiguration deformation planning problem. The problem can be summarized as follows: given an initial configuration and a goal configuration, find a sequence of module moves that will reconfigure the robots from the initial configuration to the goal configuration [3].

Based on the different module unit structures and kinematic pairs, modular self-reconfigurable robots can be classified into three types: lattice-type, chain-type, and mobile-type. In lattice-type structures, the module units mostly have regular geometric shapes, are positioned in lattice points, and reconfigure by interacting with adjacent lattice points [4,5,6,7]. However, lattice-type modular robots face challenges in generating autonomous locomotion in dynamic environments due to inherent structural constraints. For chain-type modular robots, the modules do not require regular geometric shapes. Multiple modules can connect to form hyper-redundant chain structures, which provide the system overall with mobility capabilities. There are already many research studies on chain-type robotic systems, such as [8,9,10,11,12]. The mobile-type modular robotic system, in which each single module robot has independent movement capability, moves between modules to reconfigure in a dynamic environment. Some mobile-type robotic systems were developed in the past few years, such as [13,14,15].

The problem of deforming a modular robot autonomously from an initial to a goal configuration via motion actions on a module level is referred to as the Reconfiguration Deformation Problem. Due to constraints imposed by the modules’ own structures, such as mobility degrees of freedom, number and compatibility of docking mechanisms, and physical connection constraints between multi-module units, the system has complex constraints that make reconfiguration deformation planning more than just simple path planning for mobile robots. In addition, as the number of modules increases, the number of possible configurations for the robotic system rises exponentially. Top-down optimization control methods (centralized control) were proven to be NP-complete problems [16]. A reconfiguration algorithm derived from the description of the configuration space based on extended binary trees for shape-shifting modular robots with a triangular structure is proposed in [17]; the algorithm is capable of solving the self-reconfiguration problem for modular robots with a triangular structure. One main drawback of this approach is that collision avoidance is not addressed. Ref. [18] presented an algorithm to carry out configuration decomposition iteratively by adding virtual modules and virtual connections to solve the reconfiguration problem with the SMORES robotic system. The main limitation of this method is that the mapping between nodes with virtual connections is random, without considering the shortest path issue. A hybrid particle swarm optimization and differential evolution algorithm to optimize and minimize the distance of the total movement of formation reconfiguration was proposed in [19]; however, the target assignment is static allocation and does not consider collision avoidance.

In this paper, the focus is on self-reconfiguration deformation planning of mobile-type modular reconfigurable robotic systems. A graph-based approach is utilized for representing and recognizing modular robot reconfigurations. Additionally, a hybrid reconfiguration deformation algorithm is developed to generate the reconfiguration actions that will reconfigure the robots from the initial configuration to the goal configuration.

The innovations of this paper are as follows:A graph theory-based graphical representation is utilized to describe the topology of modular robots. A distributed configuration recognition algorithm is proposed that uses the connection information between modules to identify the topological configuration of the robot system and generate a graph-based tree description;A hybrid algorithm is developed to segment configuration into multiple subconfigurations, which share maximum common subconfigurations between initial and goal configurations, and implement the distributed dynamic mapping of free vertices;A reconfiguration action generation method is proposed that utilizes the module mapping relationship in order to achieve reconfiguration.

The remainder of this paper is organized as follows. Section 2 provides an introduction to the hardware platform and simulation model. The graph-based representation and configuration recognition algorithm are explained in Section 3. Section 4 then presents the reconfiguration planning algorithm. Next, Section 5 shows the simulation and results. Section 6 shows the simulation result analysis. Section 7 provides an experiment in the real world. Finally, Section 8 discusses the conclusions and potential future work.

## 2. Hardware Platform

The MMRP (Modular Mobile Reconfigurable Platform) system [20] is a modular robotic system where each module is a complete mobile tracked robot with four connected arms as shown in Figure 1, containing onboard sensors, an STM32 microcontroller, a UWB positioning unit, and a 2.4 GHz NRF24L01 transceiver unit.

A cam–ball-type connection mechanism as shown in Figure 2, comprising an active connector and a passive connector, performs docking and undocking functions between modules.

Each MMRP module contains four docking faces (Top, Bottom, Left, and Right), with two active connectors located on the Top and Right faces, as well as two passive connectors located on the Bottom and Left faces. The connected faces are able to exchange data via contact point by serial communication.

## 3. Configuration Representation and Recognition

### 3.1. Graph-Based Topology Representation

The modular reconfiguration and transformation task for modular robots can be described as follows: Given an initial configuration, each module undergoes a series of motions (including changes in position and reconfiguration of connections) to form a specified goal configuration. The research problem lies in mapping and allocating the start and end positions and connection states of each module between the initial and goal configurations. Graph-based topology representation can clearly express the compositional configuration relationships between modules in a modular robotic system. In particular, a modular robot configuration can be represented as an undirected graph G=(V,E), where V is a set of vertices of the graph representing the modules and E is a set of edges representing the connections between two modules in V. In graph theory, a tree is an undirected graph in which any two vertices are connected by exactly one path, or equivalently a connected acyclic undirected graph [21]. Due to the inherent nature of modular robots (e.g., a module cannot connect to itself to form a self-loop), it is reasonable and convenient to represent configurations as trees. When there are closed-loop structures between several modules in the initial or target configurations, one edge of the closed-loop should be converted into a virtual connection to transform the configuration into an acyclic configuration. Therefore, this paper assumes all configurations are acyclic configurations.

Once a configuration represented by a tree graph G=(V,E) is described, a special vertex $v_{0}\in V$ can be designated as the root of the tree graph G=(V,E). At this point, G=(V,E) is a rooted tree graph with $v_{0}$ as the root. In a rooted tree G=(V,E), the parent of a vertex v∈V is the vertex connected to v on the path to the root and every vertex has a unique parent except the root, which has no parent. A child of a vertex v is a vertex of which v is the parent. In a tree graph, any vertex can be designated as the root, but in order to ensure the efficiency of the reconfiguration algorithm discussed below, a special vertex that serves as the root is essential to maximize traversal efficiency, i.e., the center point of the tree graph.

**Theorem 1.** 
*The center of an acyclic graph always exists. It is a unique vertex or a unique pair of adjacent vertices such that removing that vertex (or pair of vertices) from the graph leaves a collection of components each having less than half of the vertices [22].*


Given a tree G composed of n vertices and a random vertex v of G, G−v is the set of all rest components after removing v from the tree G. According to Theorem 1, v is the center vertex (root) if and only if G−v has no components composed of vertex numbers more than n/2.

Each MMRP robotic module has four docking faces. Connecting two unit modules via different faces results in different system configurations. Therefore, the corresponding docking face relationships between connected unit modules must also be considered.

**Definition 1.** *A connection between a parent vertex*  u*’s connector face* $F_{u}$ *and its child vertex *v*’s connector face* $F_{v}$ *is defined as follows:*

(1)
$$\mathrm{connect}(u\;F_{u},v\;F_{v})$$

*For vertex* v*, and form the perspective of* v*, its parent, which is connected via* v*’s connector face,* F *is denoted as*  ${\overset{\leftarrow }{v}}^{F}$ *and the mating connector face of* ${\overset{\leftarrow }{v}}^{F}$ *is* $\overset{\leftarrow }{F}$.

Figure 3 shows an example of the connection between $v_{2}$ and its parent vertex $v_{0}$. The connection shown in Figure 3 is connect(0L, 2R). From the perspective of vertex $v_{2}$, its parent vertex connected via the $v_{2}$’s R face should be denoted as ${\overset{\leftarrow }{v}}_{2}{}^{R}$, which corresponds to vertex $v_{0}$ in the global view.

Since the MMRP modules have heterogeneous docking mechanisms requiring an active docking face to connect with a passive face, there are only four viable connection arrangements: connect(uT,vB), connect(uL,vR), connect(uT,vL), and connect(uR,vB) between modules u and v. Connections connect(uT,vL) and connect(uR,vB) result in motion self-locking between the two connected modules. Therefore, in the MMRP system, connections are only available between the Top and Bottom faces and between the Left and Right faces.

### 3.2. Configuration Recognition

After multiple modules are connected to form the initial configuration, the current topological configuration needs to be automatically identified, and a graph representation of the current configuration should be generated to facilitate subsequent reconfiguration planning. The modular robots connect to each other via docking mechanisms to form a tree graph topological structure. The surface of the docking mechanisms integrates contact point serial communication units, enabling half-duplex data interaction between adjacent parent–child modules. The parent–child relationships form the basis of the tree graph topological structure. By determining the parent–child relationships between adjacent modules and their corresponding connection relationships, the positions and orientations of the modules can be identified, allowing recognition and representation of the overall topological configuration. A configuration recognition algorithm that utilizes the distributed connection information between modules is proposed.

The input to this algorithm is all the modules comprising the current configuration. The output is the connection relationships between the modules in the current configuration. The basic idea is to use breadth-first search order starting from any non-leaf vertex module in the configuration to iteratively identify the parent–child connection relationships layer by layer, successfully recognizing the configuration when all leaf vertices are traversed. This configuration recognition progress can be finished in time $O(\left|V\right|)$. The flowchart is shown in Figure 4.

All modules in the current configuration simultaneously start detecting the connection status of their own connection faces. Modules with only one connected face are identified as leaf modules. The communication ports of all connected faces are switched to Receiving Mode. Among non-leaf modules, one module is randomly selected to start sending configuration identification information, including its own ID and the ID of its current connected face. For a module with n total connected faces, after n−1 connected faces receive the information, the remaining unreceived connection face switches to Sending Mode to continue transmitting the information. After all leaf modules receive the data, each module has collected the parent–child connection relationships and corresponding connection face information of the modules in the configuration. This information can be uploaded to the host computer to construct a graphical representation of the current configuration. Compared to the configuration recognition algorithm designed in [11], this algorithm only needs to detect the traversal results of leaf vertices to determine the convergence state of the configuration recognition process, which improves the efficiency of detecting the convergence state.

As discussed in Section 3.1, when a configuration is described as a tree graph, this tree graph must have one (or a pair) of central vertices serving as the root of the tree. A distributed method is proposed in [23] using the connection information between adjacent modules to find the root of a tree graph. This method requires coordinating the information transmission sequence between modules, and each module needs to complete a full graph traversal to obtain the number of all its child modules. For configurations with a large number of modules, the efficiency of this algorithm will be greatly reduced. Therefore, a centralized algorithm is adopted that sequentially removes non-leaf vertices from the tree graph on the PC side, calculates the number of vertices in each acyclic subgraph of the remaining graph, and locates the root node meeting the criteria.

Once the configuration graph G=(V,E) and its root node τ are determined, the parent–child relationships between any node v in the configuration will also be determined. The configuration can be viewed as the result of iterative growth from the root node τ layer-by-layer until the leaf vertices, as shown in Figure 5. In a rooted tree, the height of a vertex is defined as the length of the longest downward path from that vertex to a leaf. The root τ has the maximum height equivalent to the height of the tree, while leaf vertices have a height of 0.

## 4. Reconfiguration Planning Algorithm

According to the configuration identification method described in Section 3.2, the current and target configurations can be represented as $G_{c}=(V_{c},E_{c})$ and $G_{g}=(V_{g},E_{g})$, respectively. The first step in configuration matching is to determine whether the goal configuration can be reconfigured from the current configuration. For the robotic system in this paper, each unit module has independent mobility, so it only needs to be determined that the goal configuration has the same number of unit modules as the current configuration, i.e., $V_{g}=V_{c}$. After confirming the configuration reconfiguring requirement is met, further configuration reconfiguration can be performed.

There are various metrics to measure the efficiency of configuration reconfiguration strategies, such as minimizing the total movement distance of all modules or minimizing the number of reconnections. In practical applications, the docking process often involves time-consuming behaviors such as alignment correction and operating the connection mechanisms. Therefore, the reconfiguration strategy aims to maximize the reduction in reconnections between modules as much as possible, thereby improving overall reconfiguration efficiency. The core idea of our proposed reconfiguration algorithm is thus finding the maximum common subgraph (MCS) isomorphism between the initial and goal configurations, reducing the number of modules involved in docking operations during reconfiguration.

**Definition 2.** *Given two graph-based representations of modular robot configurations* $G_{c}=(V_{c},E_{c})$ *and* $G_{g}=(V_{g},E_{g})$ *, common subconfiguration is a set of graphs where* $G_{c}{}^{\prime }=(V_{c}{}^{\prime },E_{c}{}^{\prime })\subseteq G_{c}$*,* $G_{g}{}^{\prime }=(V_{g}{}^{\prime },E_{g}{}^{\prime })\subseteq G_{g}$ *such that*$G_{c}{}^{\prime }$ *and*  $G_{g}{}^{\prime }$ *are isomorphic. The corresponding bijective mapping of vertices and edges is defined as* $f^{\prime }:(V_{c}{}^{\prime },E_{c}{}^{\prime })\to (V_{g}{}^{\prime },E_{g}{}^{\prime })$*.*

**Definition 3.** *Given the condition that module*  $v_{c}\in V_{c}$ *of* $G_{c}=(V_{c},E_{c})$ *must be mapped to module* $v_{g}\in V_{g}$ *of* $G_{g}=(V_{g},E_{g})$*, the common subconfiguration with maximum common connections is called the Maximum Common Subconfiguration, with respect to* $v_{c}$ *and* $v_{g}$ *denoted as* $MCS(v_{c},v_{g})$ *with mapping* $f:({\bar{V}}_{c},{\bar{E}}_{c})\to ({\bar{V}}_{g},{\bar{E}}_{g})$ *where* ${\bar{V}}_{c}\subseteq V_{c}$ *and* ${\bar{V}}_{g}\subseteq V_{g}$*.*

To find the MCS, a DFS search algorithm is utilized, starting from the root to sequentially compare the docking face information of connected modules. This obtains the $MCS(v_{1},v_{2})$ between $G_{c}$ and $G_{g}$ relative to $v_{1}$ and $v_{2}$. Figure 6 shows an example of the common subconfiguration and MCS between two configurations.

### 4.1. Configuration Decomposition

When modular robots perform reconfiguration tasks, to reduce the number of modules involving docking operations to ensure reconfiguration efficiency, the initial and goal configurations need to be decomposed. The configurations are broken down into a maximum common subgraph with the most shared connection relationships between the initial and target configurations, along with several remaining subgraphs.

For a modular robot performing a reconfiguration task, let the initial and goal configurations be $G_{c}=(V_{c},E_{c})$ and $G_{g}=(V_{g},E_{g})$, respectively, with root modules $\tau_{c}$ and $\tau_{g}$. The maximum common subgraph of $G_{c}$ and $G_{g}$, relative to $\tau_{c}$ and $\tau_{g}$ $MCS(\tau_{c},\tau_{g})$ under the mapping $f:({\bar{V}}_{c},{\bar{E}}_{c})\to ({\bar{V}}_{g},{\bar{E}}_{g})$ where ${\bar{V}}_{c}\subseteq V_{c}$ and ${\bar{V}}_{g}\subseteq V_{g}$ can be efficiently found, and denoted as ${\bar{G}}_{c}=({\bar{V}}_{c},{\bar{E}}_{c})$. Subtracting the MCS subconfiguration ${\bar{G}}_{c}$ from the initial configuration $G_{c}$ will generate a set of disconnected subgraphs ${\tilde{G}}_{c}=\left\{\left.{\tilde{G}}_{c}^{i}=({\tilde{V}}_{c}^{},{\tilde{E}}_{c}^{})|i=1,2,\ldots n\right\}\right.$ and potentially a set of detached free module vertices ${\hat{V}}_{c}=\left\{{\hat{V}}_{c}^{j}|j=1,2,\ldots m\right\}$, which were leaf nodes in $G_{c}$ exactly. Similar operations can be applied to the goal configuration $G_{g}$ to generate ${\tilde{G}}_{g}^{\alpha }$ and potentially ${\hat{V}}_{g}^{\beta }$. This process is defined as a configuration decomposition between $G_{c}=(V_{c},E_{c})$ and $G_{g}=(V_{g},E_{g})$ with respect to the root module vertex $\tau_{c}$ and $\tau_{g}$, denoted as $CS(G_{c},\tau_{c},G_{g},\tau_{g})$. The progress of configuration decomposition can be finished in time $O({\left|V_{c}\right|}^{2})$. The process is shown in Figure 7.

### 4.2. Matching and Mapping between Subconfigurations

After decomposition of the initial configuration $G_{c}=(V_{c},E_{c})$, it consists of ${\bar{G}}_{c}=({\bar{V}}_{c},{\bar{E}}_{c})$, ${\tilde{G}}_{c}^{i}$, and ${\hat{V}}_{c}^{j}$ (suppose free vertices are present). Similarly, the goal configuration $G_{g}=(V_{g},E_{g})$ consists of ${\bar{G}}_{g}=({\bar{V}}_{g},{\bar{E}}_{g})$, ${\tilde{G}}_{g}^{\alpha }$, and ${\hat{V}}_{g}^{\beta }$ after decomposition. All the modules in ${\bar{G}}_{c}$ and ${\bar{G}}_{g}$ belong to the $MCS(\tau_{c},\tau_{g})$, and have a one-to-one mapping relationship between them under mapping $f:({\bar{V}}_{c},{\bar{E}}_{c})\to ({\bar{V}}_{g},{\bar{E}}_{g})$. For ${\tilde{G}}_{c}=\left\{\left.{\tilde{G}}_{c}^{i}=({\tilde{V}}_{c}^{},{\tilde{E}}_{c}^{})|i=1,2,\ldots n\right\}\right.$ composed of multiple subconfigurations, each subconfiguration ${\tilde{G}}_{c}^{i}$ can be expressed as a tree rooted with central node ${\tilde{\tau }}_{g}^{\alpha }$. Similarly, each ${\tilde{G}}_{g}^{\alpha }$ can be expressed as a tree rooted with ${\tilde{\tau }}_{g}^{\alpha }$. Therefore, the isomorphic parts between the subconfigurations of the initial and goal configurations can be computed by sequentially finding the $MCS({\tilde{\tau }}_{c}^{i},{\tilde{\tau }}_{g}^{\alpha })$, completing the relational mapping of the modules comprising the isomorphic graphs.

However, if $MCS({\tilde{\tau }}_{c}^{i},{\tilde{\tau }}_{g}^{\alpha })$ is searched sequentially between ${\tilde{G}}_{c}^{i}$ and ${\tilde{G}}_{g}^{\alpha }$, it will require i×α attempts. This may also find a large number of isomorphic mappings consisting of only a small number of nodes, which is not very meaningful for overall configuration mapping and will greatly reduce reconfiguration planning efficiency. Therefore, a height-adjacent matching search rule is proposed to reduce the number of search attempts.

Subconfigurations eligible for MCS search must satisfy the condition $\left|h({\tilde{\tau }}_{c}^{i})-h({\tilde{\tau }}_{g}^{\alpha })\right|\leq 1$.

The height-adjacent matching search rule states that for subconfiguration ${\tilde{G}}_{c}^{i}$ in the current configuration with central node ${\tilde{\tau }}_{c}^{i}$ as its root, having height $h({\tilde{\tau }}_{c}^{i})$, ${\tilde{G}}_{c}^{i}$ can only search MCS with subconfigurations ${\tilde{G}}_{g}^{\alpha }$ that have either the same height as ${\tilde{G}}_{c}^{i}$ or are within 1 layer of height difference, as shown in Figure 8.

Configuration matching starts with the subconfiguration ${\tilde{G}}_{c}^{k}$ with the highest level in the current configuration’s set of subconfigurations ${\tilde{G}}_{c}^{}$. ${\tilde{G}}_{c}^{k}$ performs isomorphic mapping with the subconfiguration ${\tilde{G}}_{g}^{\gamma }$ from the goal configuration’s set ${\tilde{G}}_{g}^{}$ that meets the search rule, $MCS({\tilde{\tau }}_{c}^{k},{\tilde{\tau }}_{g}^{\gamma })$ under mapping $f:({\tilde{V}}_{c},{\tilde{E}}_{c})\to ({\tilde{V}}_{g},{\tilde{E}}_{g})$ can be computed. This forms a new pair of isomorphic subconfigurations denoted as {${\bar{G}}_{c}^{k}$, ${\bar{G}}_{g}^{\gamma }$}. Then, the configuration decomposition process in Section 4.1 is repeated as $CS({\tilde{G}}_{c}^{k},{\tilde{\tau }}_{c}^{k},{\tilde{G}}_{g}^{\gamma },{\tilde{\tau }}_{g}^{\gamma })$, generating new subconfigurations ${\tilde{G}}_{c}^{k}{}^{\prime }$, ${\tilde{G}}_{g}^{\gamma }{}^{\prime }$ and free vertices ${\hat{V}}_{c}^{k}$, ${\hat{V}}_{g}^{\gamma }$. The newly generated ${\tilde{G}}_{c}^{k}{}^{\prime }$ is added to set ${\tilde{G}}_{g}^{}$ while ${\hat{V}}_{c}^{k}$ is added to set ${\hat{V}}_{c}$ to participate in subsequent subconfiguration matching and mapping, and ${\tilde{G}}_{g}^{\gamma }{}^{\prime }$, ${\hat{V}}_{g}^{\gamma }$ likewise. For the worst conditions, this progress can be finished in time $O({\left|{\tilde{V}}_{c}\right|}^{3})$. However, in reality, due to the height-adjacent matching search rule, this progress should be much smaller than the worst.

Isomorphic configuration searching and mapping between subconfigurations is completed by iterating this process until every subconfiguration in ${\tilde{G}}_{c}$ or ${\tilde{G}}_{g}$ has been visited. At this point, if there are still unmatched subconfigurations existing, the connections between modules in ${\tilde{G}}_{c}$ are disconnected to make them free vertices, which are all merged into set ${\hat{V}}_{c}$, and ${\tilde{G}}_{g}$ likewise. It is worth noting that at this point the number of vertices in set ${\hat{V}}_{c}$ equals the number in set ${\hat{V}}_{g}$. After this, the successfully matched and mapped subconfigurations ${\bar{G}}_{c}^{k}$ are collectively moved to the corresponding ${\bar{G}}_{g}^{\gamma }$ locations to await subsequent steps.

### 4.3. Distributed Dynamic Mapping of Free Vertices

At this point, the current and goal configurations will consist of several isomorphic subconfigurations and free vertices, as shown in Figure 9a.

By establishing a one-to-one mapping between the free vertices in ${\hat{V}}_{c}$ and ${\hat{V}}_{g}$, the overall reconfiguration mapping plan from the initial configuration to the goal configuration can be completed. This process must consider not only the travel efficiency of moving the current free vertex modules to the target locations, but also the computational load, which grows rapidly when there are a large number of free vertex modules. Continuing centralized planning would reduce overall planning efficiency. Therefore, for this stage of reconfiguration planning, a distributed dynamic mapping strategy using artificial forces is proposed to solve the mapping problem from free vertices ${\hat{V}}_{c}$ to target vertices ${\hat{V}}_{g}$ dynamically in a distributed way.

For vertices ${\hat{v}}_{g}\in {\hat{V}}_{g}$, there exists a connection between ${\hat{v}}_{g}$ and a subconfiguration ${\bar{G}}_{g}^{\alpha }$ in the goal configuration that has completed subconfiguration mapping, and denoted as connect(${\hat{v}}_{g}$,${\bar{v}}_{g}^{\alpha }$) where ${\hat{v}}_{g}\in {\hat{V}}_{g}$ and ${\bar{v}}_{g}^{\alpha }\in {\bar{G}}_{g}^{\alpha }$, as shown in Figure 9b. Each vertex in ${\hat{V}}_{g}$ has at least one such connection, which determines the position of the vertex in ${\hat{V}}_{g}$ within the goal configuration. Since ${\bar{v}}_{g}^{\alpha }$ exists in the MCS, the target positions of vertices in ${\hat{V}}_{c}$ are known. The target positions are broadcast to all vertex in ${\hat{V}}_{c}$. Under an artificial forces field, the vertex in ${\hat{V}}_{c}$ automatically selects the nearest target position from its current position for matching. Since ${\hat{V}}_{c}$ and ${\hat{V}}_{g}$ have equal numbers of vertices, the vertices will ultimately form a one-to-one mapping $f:{\hat{V}}_{c}\to {\hat{V}}_{g}$. In this process, three types of artificial forces are constructed: attractive force, repulsive force, and obstacle avoidance force.

The attractive force denoted by $\vec{F_{a}}$, from the vertex in ${\hat{V}}_{c}$ towards goal position, is to attract the nearest free vertex to the position of ${\hat{V}}_{g}$ to form a mapping from ${\hat{V}}_{c}$ to ${\hat{V}}_{g}$. The force is calculated as
(2)$$\vec{F_{a}}=\frac{K_{a}}{{\vec{d_{v_{c},v_{g}}}}^{2}}$$
where $\vec{d_{v_{c},v_{g}}}=\vec{\sqrt{{\left(x_{v_{c}}-x_{v_{g}}\right)}^{2}+{\left(y_{v_{c}}-y_{v_{g}}\right)}^{2}}}$ is the Euclidean distance from the vertex in ${\hat{V}}_{c}$ towards each goal position, and $K_{a}$ is the attractive force factor.

The repulsive force denoted by $\vec{F_{r}}$ is to push two free vertices away when their distance is less than the specified value. The force is calculated as
(3)$$\vec{F_{r}}=\left\{\begin{array}{cc} \frac{K_{r}}{{\vec{d_{v_{1},v_{2}}}}^{2}} & ,\;\mathrm{if}\;0<\left|\vec{d_{v_{1},v_{2}}}\right|<d_{r}\\ 0 & else \end{array}\right.$$
where $d_{v_{1},v_{2}}$ is the Euclidean distance between any two free vertices in ${\hat{V}}_{c}$, and $K_{r}$ is a repulsive force factor.

The obstacle avoidance force denoted by $\vec{F_{obs}}$ is to push free vertices away from isomorphic subconfigurations that are mapped already. And the force is calculated as
(4)$$\vec{F_{obs}}=\left\{\begin{array}{cc} -\frac{K_{r}}{\vec{d_{obs}}}, & 0<\left|\vec{d_{obs}}\right|<d_{safe-}\\ 0, & d_{safe-}\leq \left|\vec{d_{obs}}\right|\leq d_{safe+}\\ K_{r}\cdot \vec{d_{obs}}, & \left|\vec{d_{obs}}\right|>d_{safe+} \end{array}\right.$$
where $\vec{d_{obs}}$ is the Euclidean distance from the free vertex to the subconfigurations.

The attractive force $\vec{F_{a}}$ pulls the free vertex towards the nearest target point while the repulsive force $\vec{F_{r}}$ pushes free vertices away to keep a safe distance avoiding collision. The obstacle avoidance force $\vec{F_{obs}}$ keeps the free vertices moving in a ring area around the robotic system. Combing the forces above, the resultant force (shown in Figure 10) acting on the free vertex is as follows:(5)$$\vec{F}=\vec{F_{a}}+\vec{F_{r}}+\vec{F_{obs}}$$

The free vertex moves simply by the resultant force acting on it without centralized planning. When a free node reaches its target position, the target position stops providing attractive force and the free vertex stops moving. This establishes a mapping $f:{\hat{v}}_{c}\to {\hat{v}}_{g}$ between ${\hat{v}}_{c}$ and the corresponding ${\hat{v}}_{g}$ represented by the target position node.

When all free nodes have reached their target positions, at this point, all vertices in the initial configuration and vertices in the goal configuration have a one-to-one mapping $f:V_{c}\to V_{g}$. The vertices belonging to the MCS satisfy mapping $f:({V^{\prime }}_{c},{E^{\prime }}_{c})\to ({V^{\prime }}_{g},{E^{\prime }}_{g})$. This progress can be finished in time O(d), where d is the total distance of free vertices traveled.

### 4.4. Reconfiguration Action Generation based on Mapping

Once the Reconfiguration Mapping process is completed, corresponding reconfiguration actions can be determined. The mapping $f:V_{c}\to V_{g}$ provides the mapping relations that map each module and corresponding connection from the initial configuration $G_{c}=(V_{c},E_{c})$ into the goal configuration $G_{g}=(V_{g},E_{g})$. The reconfiguration actions can be generated by comparing the edge mappings of the corresponding mapped module pair between $G_{c}$ and $G_{g}$. Under the mapping $f:V_{c}\to V_{g}$, if a module pair $\left(v_{c},v_{g}\right)$ where $v_{c}\in V_{c}$ and $v_{g}\in V_{g}$ is in any MCS during the mapping process meanwhile its corresponding parent module pair $\left({\overset{\leftarrow }{v}}_{c}^{F},{\overset{\leftarrow }{v}}_{g}^{F}\right)$ is in the same MCS, indicating that no reconfiguration action is needed for this connected module pair. Otherwise, ${\overset{\leftarrow }{v}}_{g}^{F}$ will be mapped to a unique vertex ${v^{\prime }}_{c}$ (which is not the parent vertex ${\overset{\leftarrow }{v}}_{c}^{F}$ of $v_{c}$) under the mapping $f^{-1}:V_{g}\to V_{c}$, and the reconfiguration actions are undocking $v_{c}$ from ${\overset{\leftarrow }{v}}_{c}^{F}$ by removing connect($v_{c}F$, ${\overset{\leftarrow }{v}}_{c}^{F}\overset{\leftarrow }{F}$) and then docking $v_{c}$ with ${v^{\prime }}_{c}$ by establishing connect($v_{c}F$, ${v^{\prime }}_{c}F$). The reconfiguration actions will be generated by traversing all modules in $G_{c}$. This process can be carried out in time $O(\left|V_{c}\right|)$. Following the complete traversal of all modules, the initial configuration $G_{c}$ will be transformed into the goal configuration $G_{g}$ via the execution of the reconfiguration actions computed by the planning algorithm.

## 5. Simulation

The reconfiguration algorithm is implemented in Python and validated in the robotics simulator CoppeliaSimEdu. A modular robotic system composed of 9 MMRP robots is simulated to reconfigure from an initial cross configuration to a probe configuration to validate the reconfiguration deformation planning method proposed in this paper. The initial and goal configurations are shown in Figure 11.

The first step involves running configuration recognition algorithm 1 to detect all connections in the initial configuration. Each module in the configuration detects the connection status of its own faces to identify the leaf modules, which are Modules 6, 7, 8, and 9. A non-leaf Module 2 is randomly selected to start discovering all the connections, and the sequence of discovering is shown in Table 1.

The graph of initial cross configuration $G_{c}=(V_{c},E_{c})$ can be built based on the above connections and messages. And the graph of goal configuration $G_{g}=(V_{g},E_{g})$ can also be built. The initial and goal configuration topology graph-based representations are shown in Figure 12. With $G_{c}=(V_{c},E_{c})$ and $G_{g}=(V_{g},E_{g})$, the roots $\tau_{c}$ and $\tau_{g}$ can be computed. Root $\tau_{c}$ corresponds to Module 1 in $G_{c}$ and root $\tau_{g}$ corresponds to Module 1 in $G_{g}$.

According to graphs $G_{c}=(V_{c},E_{c})$ and $G_{g}=(V_{g},E_{g})$, $MCS(\tau_{c},\tau_{g})=MCS(1,1)$ contains five modules, and the mapping relations are shown in Table 2. ${\bar{G}}_{c}$ and ${\bar{G}}_{g}$ are generated.

The result of configuration decomposition $CS(G_{c},\tau_{c},G_{g},\tau_{g})=CS(G_{c},1,G_{g},1)$ is shown in Figure 13.

After performing configuration decomposition, subconfigurations were generated: subconfiguration ${\tilde{G}}_{c}^{1}$ rooted at ${\tilde{\tau }}_{c}^{1}$, corresponding to Module 4 in the initial configuration, and subconfigurations ${\tilde{G}}_{g}^{1}$ rooted at ${\tilde{\tau }}_{g}^{1}$ corresponding to Module 8 and ${\tilde{G}}_{g}^{2}$ rooted at ${\tilde{\tau }}_{g}^{2}$ corresponding to Module 5 in the goal configuration. Module 9 and Module 7, which are identified as free vertices, are added to set ${\hat{V}}_{c}^{}$. $h({\tilde{\tau }}_{c}^{1})=h({\tilde{\tau }}_{g}^{1})=h({\tilde{\tau }}_{g}^{2})=2$, which means MCS matching searching between ${\tilde{G}}_{c}^{1}$ and ${\tilde{G}}_{g}^{1}$ (or ${\tilde{G}}_{c}^{1}$ and ${\tilde{G}}_{g}^{2}$) is feasible. $MCS({\tilde{\tau }}_{c}^{1},{\tilde{\tau }}_{g}^{1})=MCS(4,8)$ contains two modules and the mapping relations are shown in Table 3. Module 5 and Module 6 undock all connected faces to convert them into free vertices. Based on the mapping result, Module 4 and Module 8 move as a whole to the corresponding position to await subsequent steps.

Free vertices Module 9 and Module 7 receive the target positions. Under the artificial potential field, they dynamically find the nearest target nodes and move to those positions and the mapping $f:{\hat{V}}_{c}\to {\hat{V}}_{g}$ is 7 → 5, 9 → 6 according to the simulation result. This process is shown in Figure 14.

At this point, one-to-one mapping relations $f:V_{c}\to V_{g}$ are established between all vertices in the initial configuration and goal configuration. The reconfiguration actions are generated as detailed in Table 4.

Once the connectors complete their assigned reconfiguration actions, the reconfiguration deformation process from the initial cross configuration into the goal probe configuration process is finished. The reconfiguration process is shown in Figure 15.

## 6. Simulation Result Analysis

In this part, a simulation result analysis and discussion of graph-based hybrid reconfiguration deformation planning are presented. In comparison to the virtual connections reconfiguration algorithm (VCR) proposed in [18], taking the cross configuration to probe configuration example in Section 5, although the configuration deformation process for both methods requires 3 docking and undocking actions, the VCR does not consider the distance to the target configuration when establishing virtual connections between modules, as all virtual connections are equivalent. This can result in discarding closer modules in favor of more distant ones, such as mapping Module 9 to Module 5 and Module 7 to Module 6, even though closer module mappings are available. The dynamic mapping method for free vertices based on potential field forces solves this problem while also addressing the multi-objective collision issue discussed in [19]. Take, for another example, the driver configuration to snake configuration case proposed in [18] as shown in Figure 16. Using the method proposed in this paper, the reconfiguration can be completed with two docking and undocking actions, compared to four actions required by the method VCR.

Via comparison with the VCR method, the algorithm proposed in this paper not only reduces the number of docking and undocking actions during reconfiguration but also resolves the multi-objective node mapping conflict issue.

From a computational complexity perspective, the most time-consuming part of the algorithm proposed in this paper is the matching and mapping between subconfigurations process, which has a time complexity of $O({\left|{\tilde{V}}_{c}\right|}^{3})$. In comparison, the VCR method has a time complexity of $O({\left|V_{c}\right|}^{3})$. Although both algorithms have the same order of complexity, since free vertices are dynamically mapped in a distributed manner, $\left|{\tilde{V}}_{c}\right|\leq \left|V_{c}\right|=\left|V_{g}\right|$ for the same configuration deformation task. In actual engineering applications, involving fewer modules in high-level planning will significantly reduce decision time, thereby decreasing the time required for configuration deformation, which has positive implications for practical engineering applications.

## 7. Experiment

To validate the effectiveness of the graph-based hybrid configuration deformation planning proposed in this paper, an experiment was conducted in the real world. The experiment demonstrated a configuration deformation from a Series configuration to a T configuration using 4 MMRP robots. The modular robots communicate with the PC via NRF24L01 for data transmission. Relative position and pose relationships between modules are obtained using UWB positioning units.

The initial and goal graph representations are shown in Figure 17 where $\tau_{c}$ is Module 2 and $\tau_{g}$ is Module 1, respectively.

After Configuration Decomposition progress, $MCS(\tau_{c},\tau_{g})=MCS(2,1)$ contains two modules under mapping 2 → 1 and 1 → 2. Under the influence of artificial forces, the free vertices Module 3 and 4 dynamically search for the nearest target positions and eventually form the mappings: 3 → 3 and 4 → 4. The experiment of configuration deformation from Series configuration into T configuration is shown in Figure 18.

It should be noted that due to the need for further improvements in the current connection mechanism design and the limited number of fabricated connection arms, the angles of the connection rods relative to Modules 2 and 3 were manually adjusted in the experiment to represent different docking faces on the same module. The docking faces represented by the connection rods are marked in the figures.

Experiment results verify the effectiveness of the proposed algorithm. The main shortage discovered during the experiment was problems with the docking mechanism design. Since the MMRP platforms lack lateral mobility, the lateral docking between two modules requires manual adjustment of the relative angle between docking faces. Therefore, the optimization of the docking mechanism will be considered in future work. One proposed improvement is to add a radial joint to the connection arms, so that the active and passive faces can rotate 90° and −90°, respectively during lateral docking between two modules, enabling lateral docking functionality.

## 8. Conclusions

In this paper, a new reconfiguration deformation planning for modular robots is presented. Graph-based representations are utilized to model and recognize the topological configuration of the modules based on distributed information. An available hybrid algorithm is developed to segment configuration into multiple subconfigurations, which share maximum common subconfigurations between initial and goal configurations and implement the distributed dynamic mapping of free vertices. The module mapping relationship between the initial and target configurations is utilized to generate reconfiguration actions to achieve reconfiguration between the configurations. Simulation and experiment verify the effectiveness of this algorithm. The reconfiguration deformation planning algorithm proposed in this paper has universal applicability across mobile-type modular self-reconfigurable robotic systems.

There are still several issues in this research that need to be further addressed in future work, such as optimizing the physical structure to resolve lateral docking limitations, coordinated motion control of multiple modules, and practical configuration planning for real-world engineering applications. Future work will focus on mechanical structure optimization, coordinated motion control, and navigation of connected modular robots in dynamic environments. Improving the algorithm’s capability in handling Hamiltonian cycles in topological configurations will also be a direction for future research.

## Figures and Tables

**Figure 1 sensors-23-07892-f001:**
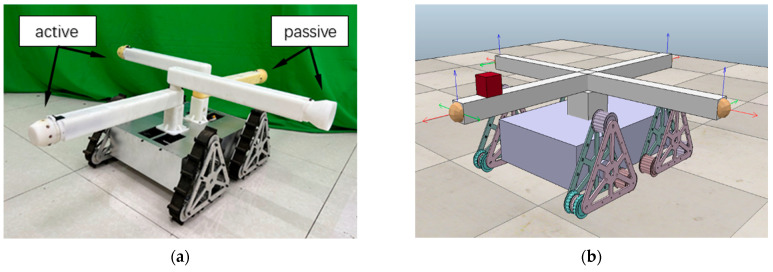
(**a**) An MMRP module with four docking faces—two active and two passive docking faces; (**b**) MMRP module simulated model in CoppeliaSim Edu (Version 4.1.0).

**Figure 2 sensors-23-07892-f002:**
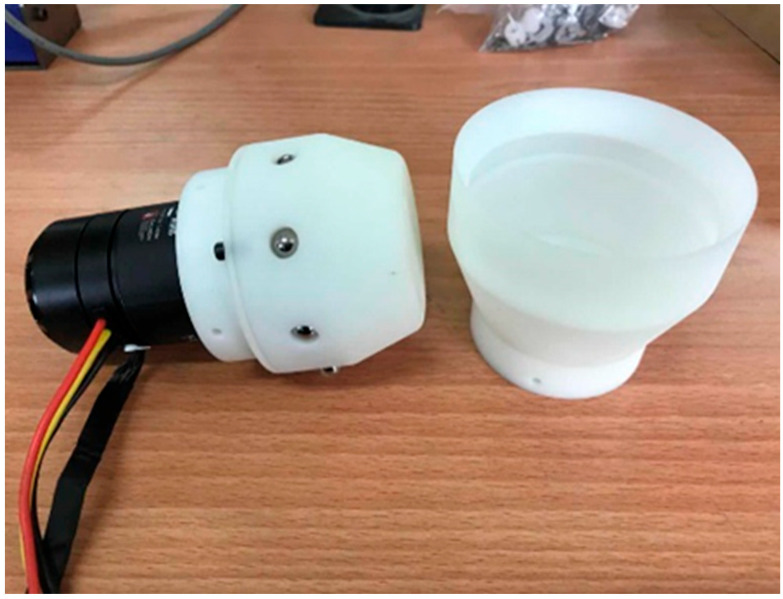
A pair of cam–ball-type connectors. Active connector on the left and passive connector on the right.

**Figure 3 sensors-23-07892-f003:**
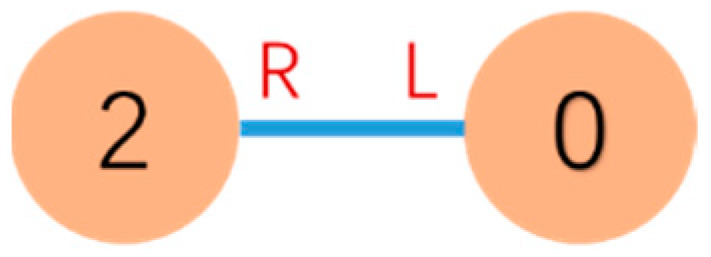
Due to Definition 1, the connection is defined as connect(0L,2R). ${\overset{\leftarrow }{v}}_{2}{}^{R}=v_{0}$ and the mating connector face of ${\overset{\leftarrow }{v}}_{2}{}^{R}$ is $\overset{\leftarrow }{L}$.

**Figure 4 sensors-23-07892-f004:**
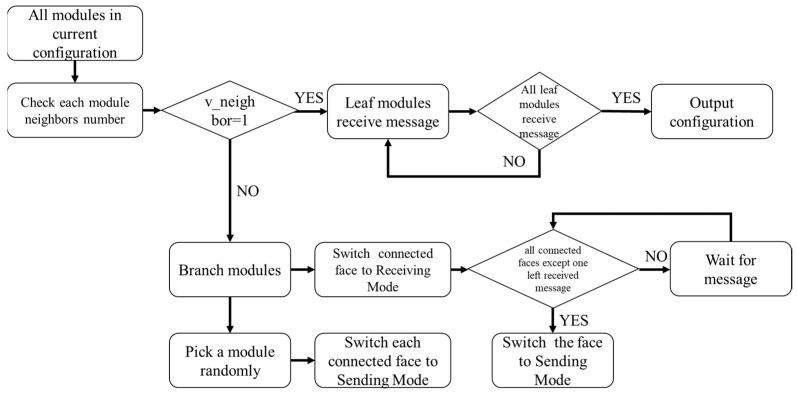
Configuration recognition flowchart.

**Figure 5 sensors-23-07892-f005:**
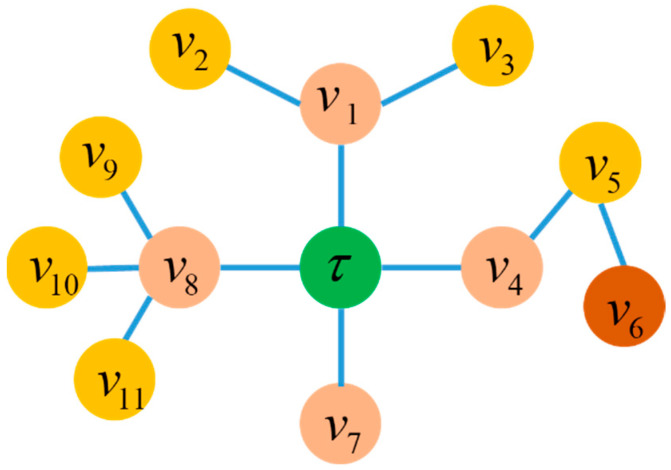
A graph-based tree G=(V,E) rooted with τ. The height of the tree is h(τ)=3.

**Figure 6 sensors-23-07892-f006:**
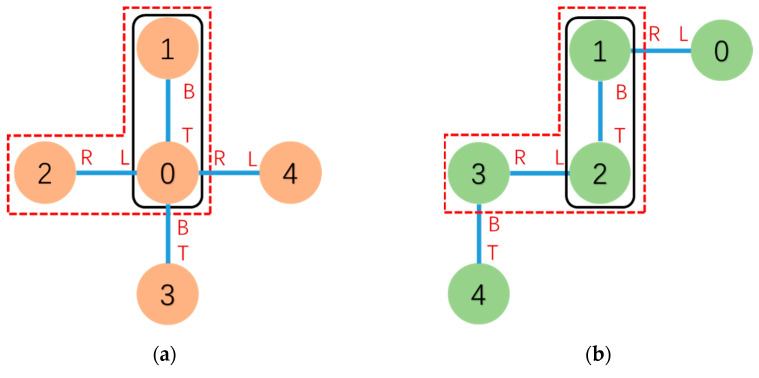
One of the common subconfigurations of (**a**,**b**) is marked by “—” with mapping 0 → 2, 1 → 1 and connect(0T,1B) → connect(2T,1B). The subgraphs of (**a**,**b**) marked by “- -” are MCS(0, 2) with mapping 0 → 2, 1 → 1, 2 → 3 and connect(0T,1B) → connect(2T,1B), connect(0L,2R) → connect(2L,3R).

**Figure 7 sensors-23-07892-f007:**
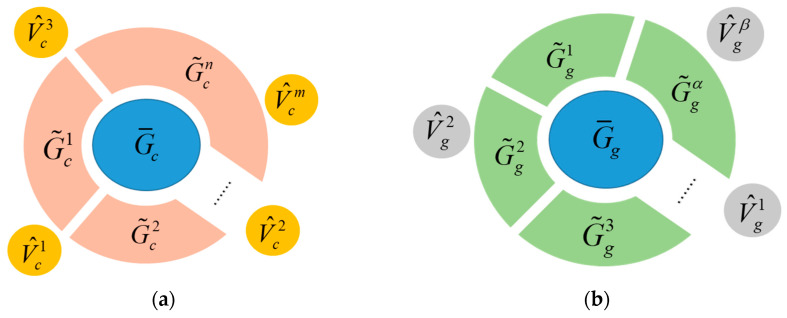
Configuration decomposition for initial configuration (**a**) and goal configuration (**b**).

**Figure 8 sensors-23-07892-f008:**
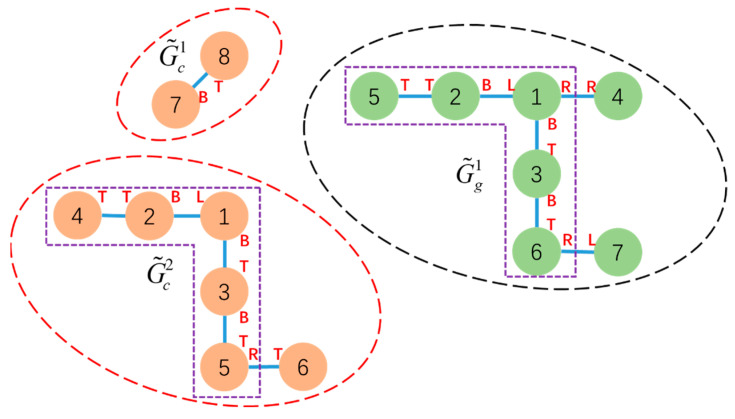
Subconfigurations for ${\tilde{G}}_{c}^{i}$ are encircled by “--” while ${\tilde{G}}_{g}^{\alpha }$ is encircled by “--”. The heights of each subconfiguration are $h({\tilde{G}}_{c}^{1})=h({\tilde{v}}_{c7})=1$, $h({\tilde{G}}_{c}^{2})=h({\tilde{v}}_{c1})=3$, and $h({\tilde{G}}_{g}^{1})=h({\tilde{v}}_{g1})=3$. The MCS searching between ${\tilde{G}}_{c}^{2}$ and ${\tilde{G}}_{g}^{1}$ is available.

**Figure 9 sensors-23-07892-f009:**
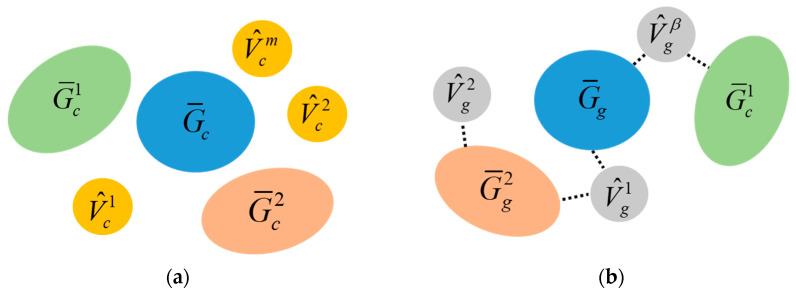
Current configuration (**a**) and goal configuration (**b**) consist of several isomorphic subconfigurations and free vertices.

**Figure 10 sensors-23-07892-f010:**
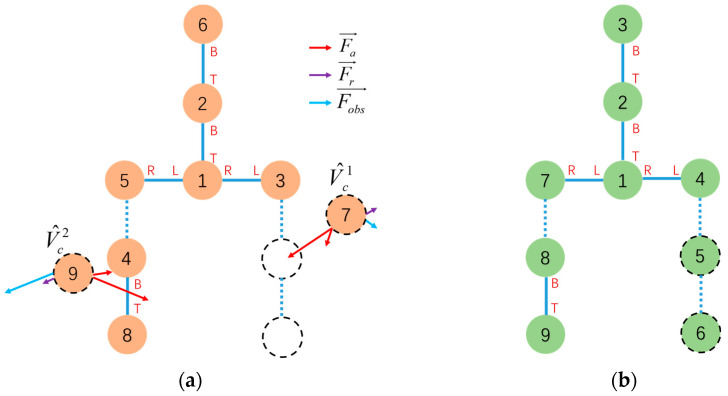
In the current configuration (**a**), the dashed circles indicate the corresponding target positions in the target configuration (**b**). The arrows show the forces acting on the free vertices, which will move the vertices to the nearest target positions and establish a mapping with the vertices in the goal configuration.

**Figure 11 sensors-23-07892-f011:**
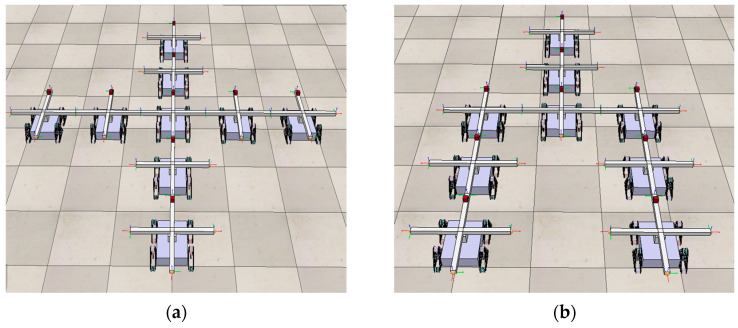
Reconfigure the initial configuration—cross configuration (**a**) into the goal configuration—probe configuration (**b**), with 9 MMRP modules involved. The connector marked in red is Top-Face.

**Figure 12 sensors-23-07892-f012:**
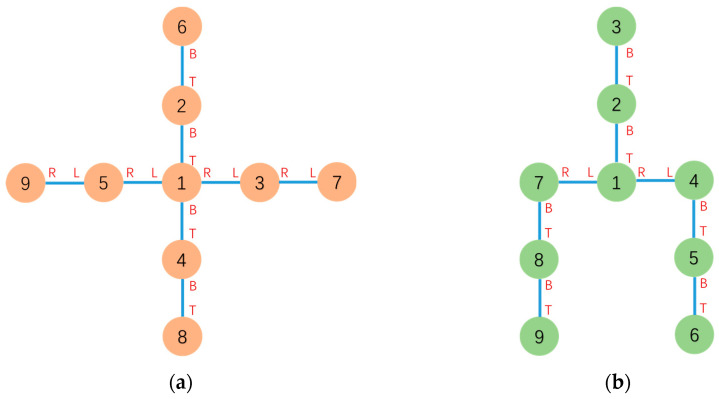
(**a**) Graph-based representation of the initial cross configuration. (**b**) Graph-based representation of the goal probe configuration.

**Figure 13 sensors-23-07892-f013:**
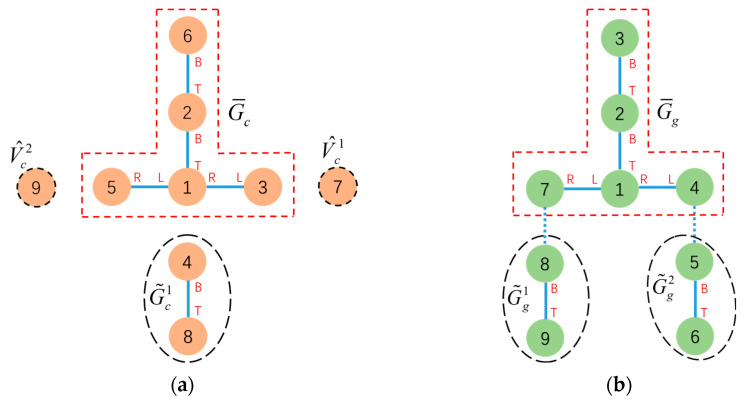
The subconfigurations and free vertex obtained after applying $CS(G_{c},1,G_{g},1)$ operations to the initial configuration (**a**) and goal configuration (**b**) is indicated by “--”. ${\bar{G}}_{c}$ and ${\bar{G}}_{g}$ are indicated by “--”.

**Figure 14 sensors-23-07892-f014:**
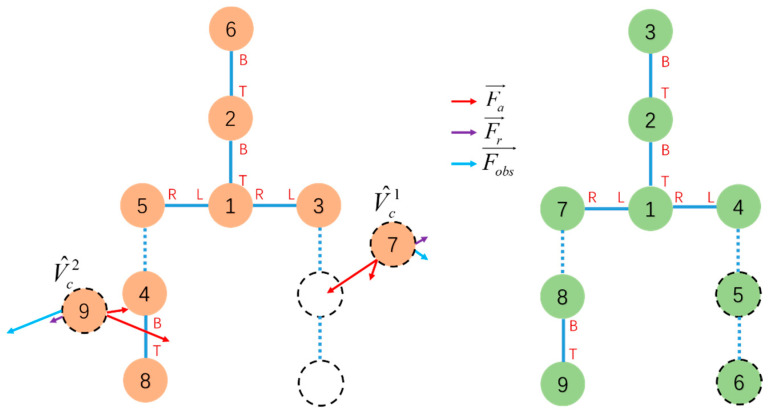
Under the influence of an artificial force field, the free vertex Module 7 and Module 9 dynamically seek out their target to complete the vertex mapping.

**Figure 15 sensors-23-07892-f015:**
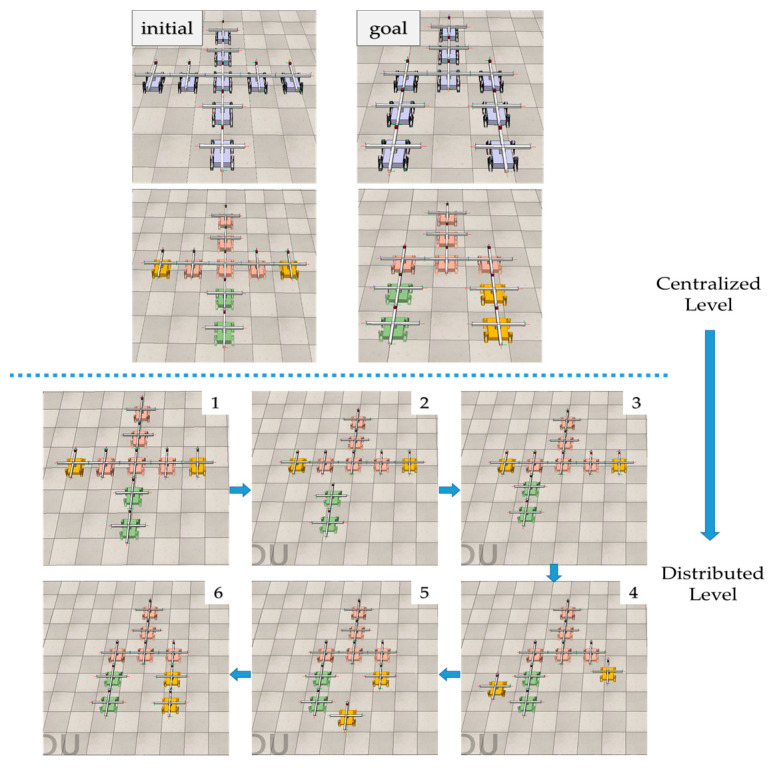
Reconfiguration process from the initial cross configuration into the goal probe configuration. Differently colored module clusters present the corresponding mapping relations between initial and goal configurations after configuration decomposition and subconfiguration mapping are computed in Python. Subconfiguration matching and free vertices distributed dynamic mapping to reconfigure goal configuration.

**Figure 16 sensors-23-07892-f016:**
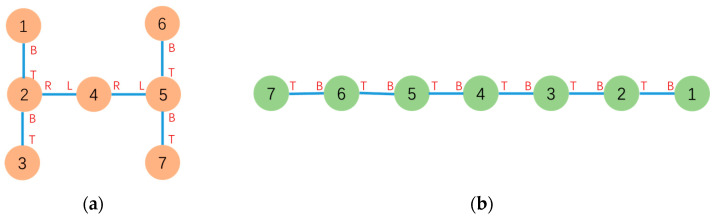
(**a**) Graph-based representation of the initial driver configuration. (**b**) Graph-based representation of the goal snake configuration.

**Figure 17 sensors-23-07892-f017:**
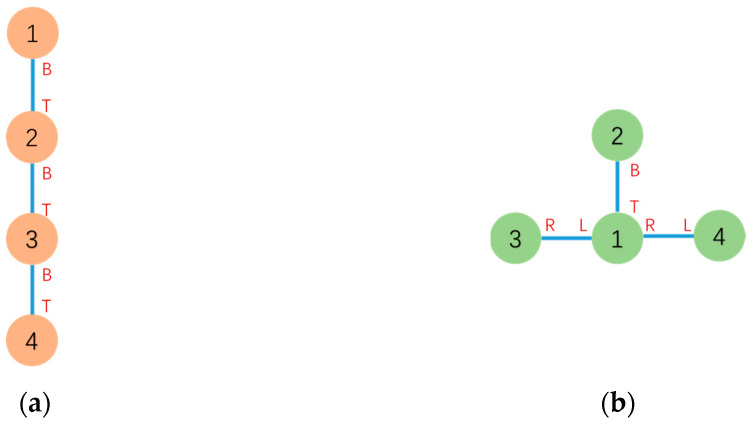
(**a**) Graph-based representation of the initial series configuration. (**b**) Graph-based representation of the goal T configuration.

**Figure 18 sensors-23-07892-f018:**
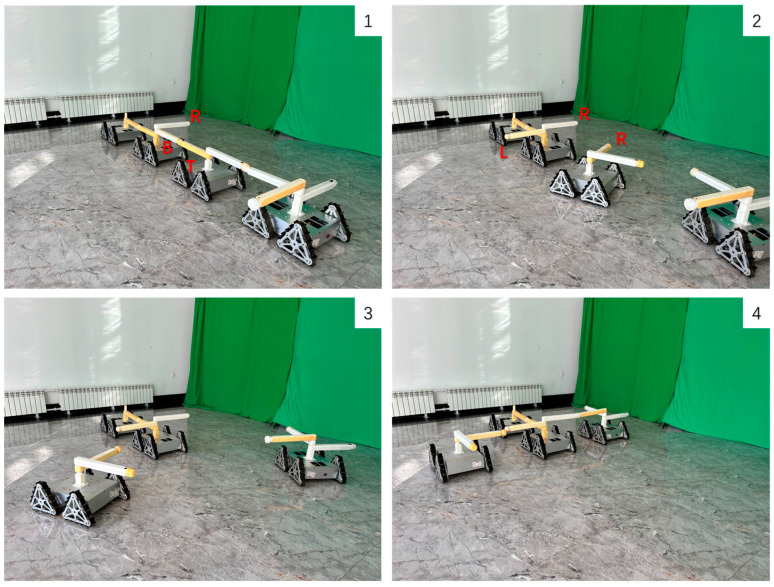
The experiment of configuration deformation from Series configuration into T configuration.

**Table 1 sensors-23-07892-t001:** Configuration recognition sequence for cross configuration.

Search Level	Connection
Level 1	connect(2T, 6B) connect(2B, 1T)
Level 2	connect(1R, 3L) connect(1B, 4T) connect(1L, 5R)
Level 3	connect(3R, 7L) connect(4B, 8T) connect(5L, 9R)

**Table 2 sensors-23-07892-t002:** Mapping relations in MCS(1, 1).

Vertex Mapping	Edge Mapping
1 → 1	$\tau_{c}$ → $\tau_{g}$
5 → 7	connect(1L, 5R) → connect(1L, 7R)
2 → 2	connect(1T, 2B) → connect(1T, 2B)
6 → 3	connect(2T, 6B) → connect(2T, 3B)
3 → 4	connect(1R, 3L) → connect(1R, 4L)

**Table 3 sensors-23-07892-t003:** Mapping relations in MCS(4, 8).

Vertex Mapping	Edge Mapping
4 → 8	${\tilde{\tau }}_{c}^{1}$→ ${\tilde{\tau }}_{g}^{1}$
8 → 9	connect(4B, 8T) → connect(8B, 9T)

**Table 4 sensors-23-07892-t004:** Reconfiguration actions for cross configuration to probe configuration.

Action	ID	Face	ID	Face
Undock	4	Top	1	Bottom
Dock	4	Top	5	Bottom
Undock	7	Left	3	Right
Dock	7	Bottom	3	Top
Undock	9	Right	5	Left
Dock	9	Top	7	Bottom

## Data Availability

Not applicable.

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
