# Peer review of "A Graph-Based Hybrid Reconfiguration Deformation Planning for Modular Robots"

_sensors, 2023, doi:10.3390/s23187892_

Round 1

Reviewer 1 Report

In this manuscript, the authors developed a graph based hybrid reconfiguration deformation planning method for modular robots. Overall, this paper is well written and the methodology is clearly described. The reviewer has the following comments.

1.       The authors are suggested to describe the suitability of submitting this work to Sensors journal. The reviewer feels that this paper might be more suitable for journals like Robotics since this paper is clearly not focused on any sensor developments.

2.       The novelty of this paper should be highlighted more clearly in the introduction section.

3.       Figure 1, it appears that only the right connector is passive in the figure, however, based on the description, there should be two passive connectors for each module.

4.       The authors should discuss the challenges of realizing this deformation planning approach using real robot modules in the future work part.

Reviewer 3 Report

Dear Sir,

I have read the present work with interest. The idea addressed is well-done and innovative. I am pleased to recommend its publications after addressing some minor points:

1-Why not citing Robotics and Autonomous Systems 147, 103930 (2022)? This published paper address a comparable issue. 

2-Why several definitions? no Lemmas, axioms, theorems.

3-Some definitions are unclear, e,g. Definitions 2 and 4.

4-In the same definition, italic and non-italic are merged.

5-Do we need all these definitions to estimate the force?

6-Any Hamiltonian cycle?

7-What about synchronization? self-reconfiguration?

Dear Sir,

I have read the present work with interest. The idea addressed is well-done and innovative. I am pleased to recommend its publications after addressing some minor points:

1-Why not citing Robotics and Autonomous Systems 147, 103930 (2022)? This published paper address a comparable issue. 

2-Why several definitions? no Lemmas, axioms, theorems.

3-Some definitions are unclear, e,g. Definitions 2 and 4.

4-In the same definition, italic and non-italic are merged.

5-Do we need all these definitions to estimate the force?

6-Any Hamiltonian cycle?

7-What about synchronization? self-reconfiguration?

Reviewer 4 Report

The paper addresses the complex challenge of reconfiguration planning in self-reconfigurable modular robots through an innovative approach. By integrating graph-based representations, sub configuration commonality, and dynamic mapping, the authors propose a hybrid algorithm that efficiently plans reconfigurations between initial and goal configurations.

The significance of this work lies in its relevance to the field of self reconfigurable robotics, where the ability to adapt and alter shape is of great importance. The authors effectively outline the complexities associated with the problem due to module constraints and exponential configuration possibilities.

The proposed hybrid algorithm is a notable contribution. By utilizing graph-based representations, the algorithm efficiently recognizes module configurations. The emphasis on identifying shared sub configurations between initial and goal states simplifies the problem and enhances planning efficiency. The concept of utilizing module mapping relationships further adds to the novelty of the approach.

Simulation results provide empirical evidence of the algorithm's efficacy in achieving reconfiguration. However, to enhance the paper's impact, a comparative analysis with existing methods could underscore its advantages. Additionally, addressing real-world implementation challenges and discussing the algorithm's computational complexity would provide a more comprehensive perspective.

Nevertheless, the content of the paper might be more aligned with journals such as "Machines" or "Applied Sciences" rather than a „Sensors” journal. Additionally, it's worth noting that some of the references provided in the paper appear to be slightly outdated. While the paper presents the proposed algorithm effectively, a comparison with existing approaches in the literature and a more comprehensive discussion of the algorithm's advantages over other methods would enhance the paper's impact.

While the simulation results are valuable, discussing potential challenges and considerations for implementing the proposed algorithm in real-world scenarios, including hardware limitations and environmental constraints, would provide a more holistic view.

Providing insights into the computational complexity of the proposed hybrid algorithm would help readers understand the algorithm's efficiency and scalability in different scenarios.

In conclusion, the research is a valuable contribution to adaptable robotics. The paper's strengths lie in its innovative methodology, problem relevance, and clear presentation. Further improvements could involve deeper contextualization, comparative analysis, and insights into practical implementation considerations. Overall, the paper offers valuable insights for advancing the capabilities of self-reconfigurable robotic systems.

Author Response

We would like to thank you for taking the time and effort to provide insightful feedback on our manuscript.  Please see the attachment.

Round 2

Reviewer 2 Report

Agree to publish